# Peptaibol Production and Characterization from *Trichoderma asperellum* and Their Action as Biofungicide

**DOI:** 10.3390/jof8101037

**Published:** 2022-09-29

**Authors:** Pamela Alfaro-Vargas, Alisson Bastos-Salas, Rodrigo Muñoz-Arrieta, Reinaldo Pereira-Reyes, Mauricio Redondo-Solano, Julián Fernández, Aníbal Mora-Villalobos, José Pablo López-Gómez

**Affiliations:** 1National Center for Biotechnological Innovations, National Center for High Technology, San Jose 1174-1200, Costa Rica; 2Faculty of Microbiology, University of Costa Rica, Rodrigo Facio University City, San Jose 11501-2060, Costa Rica; 3National Nanotechnology Laboratory, National Center for High Technology, San Jose 1174-1200, Costa Rica; 4Research Center for Tropical Diseases (CIET) and Food Microbiology Research and Training Laboratory (LIMA), Faculty of Microbiology, University of Costa Rica, Rodrigo Facio University City, San Jose 11501-2060, Costa Rica; 5Instituto Clodomiro Picado, Faculty of Microbiology, University of Costa Rica, San Jose 11501-2060, Costa Rica; 6Microbiome Biotechnology Department, Leibniz Institute for Agricultural Engineering and Bioeconomy (ATB), 14469 Potsdam, Germany

**Keywords:** fermentation, biocontrol, optimization, scanning electron microscope, phytopathogenic fungi

## Abstract

Peptaibols (P_aib_), are a class of biologically active peptides isolated from soil, fungi and molds, which have interesting properties as antimicrobial agents. P_aib_ production was optimized in flasks by adding sucrose as a carbon source, 2-aminoisobutyric acid (Aib) as an additive amino acid, and *F. oxysporum* cell debris as an elicitor. P_aib_ were purified, sequenced and identified by High-performance liquid chromatography (HPLC)coupled to mass spectrometry. Afterward, a P_aib_ extract was obtained from the optimized fermentations. The biological activity of these extracts was evaluated using in vitro and in vivo methods. The extract inhibited the growth of specific plant pathogens, and it showed inhibition rates similar to those from commercially available fungicides. Growth inhibition rates were 92.2, 74.2, 58.4 and 36.2% against *Colletotrichum gloeosporioides*, *Botrytis cinerea*, *Alternaria alternata* and *Fusarium oxysporum*, respectively. Furthermore, the antifungal activity was tested in tomatoes inoculated with *A. alternata*, the incidence of the disease in tomatoes treated with the extract was 0%, while the untreated fruit showed a 92.5% incidence of infection Scanning electron microscopy images showed structural differences between the fungi treated with or without P_aib_. The most visual alterations were sunk and shriveled morphology in spores, while the hyphae appeared to be fractured, rough and dehydrated.

## 1. Introduction

Phytopathogenic fungi cause plant diseases that manifest as disfigurement, wilting, blotches and rotted tissue. These signs reduce commercial value and generate large losses in agricultural production [1,2]. The traditional strategy for fungal control consists of the application of synthetic fungicides and pesticides [1,3]. These techniques cause an imbalance in the ecosystem, affect the environment, threaten human health and increase production costs [4,5,6]. It has been argued that replacing chemical agents with eco-friendly methodologies, such as biological control or biofungicides, could bring benefits to the agricultural industry [7,8,9].

Peptaibols (P_aib_) are a large family of bioactive peptides (more than 440) composed of 7 to 20 amino acid residues (linear or cyclic) [10,11,12]. P_aib_ are characterized by the presence of a high proportion of Aib, an acetate or acyl group in the N-terminal residue and a C-terminal amino alcohol [12,13,14]. These peptides are assembled by a multi-enzyme complex called non-ribosomal peptide synthetases (NRPSs), which allow the incorporation of non-proteinogenic amino acids, such as Aib [10,15,16,17]. Their amphipathic nature allows the formation of permanent transmembrane pores that causes the exchange of cytoplasmic material and eventual cell death [11,14].

The bioactivity of P_aib_ against parasites, viruses, bacteria and pathogenic fungi has previously been reported [10,12,18,19]. In addition, its bioactivity has been proved in therapies against cancer, Alzheimer’s disease, and some human and animal diseases, thanks to their antifungal, antitrypanosomal and anthelmintic activity [11,13,14,15,20,21]. The activity of P_aib_ has already been tested in vitro against some plant pathogens, such as *Fusarium oxysporum*, *Botrytis cinerea*, *Rhizoctonia solani*, *Bipolaris sorokiniana*, *Colletotrichum lagenarium*, *Aspergillus niger*, *Sclerotium cepivorum*, *Mucor ramannianus*, *Moniliophthora perniciosa* and *Pseudomonas syringae* pv. *Lachrymans* [1,12,13,14,15,22].

*Trichoderma* is one of the most isolated and studied ascomycetes due to its agro-industrial importance as a biocontrol organism and producer of secondary metabolites with biological activity [23,24,25]. These fungi act as antagonistic parasites against plant pathogens by inducing resistance, antibiosis, mycoparasitism and competition, protecting the plant from diseases [26,27,28]. 

As an antibiosis strategy, *Trichoderma* species are well known P_aib_ producers. Some of the P_aib_ produced by *Trichoderma* species include asperelines, alamethicins, trichokonins, trichorovins, trichotoxins, trilongins, brevicelsins, etc., but the production of more than 190 of these peptides has already been reported [2,13,14,27,29,30,31,32,33]. Specifically, *T. asperellum* is an efficient peptaibol producer. This species produces at least 38 asperelines and 5 trichotoxins with verified antifungal activity [19,34]. Its efficient P_aib_ productivity and easy laboratory handling makes *T. asperellum* a good candidate for bulk production of these peptides.

Some *Trichoderma* strains are currently being used, and even commercialized, as biocontrollers because of their antimicrobial properties [6,9,29,34,35,36]. *T. harzianum* and *T. koningii* strains are marketed in Europe and North America as biocontrollers due to the action of their P_aib_ [15]. However, the whole microorganism is commercialized, not just the active compound (P_aib_). Optimizing the production and isolation of P_aib_ is critical when only the pure active component is required, such as in biomedical applications e.g., the treatment of cancer or Alzheimer’s disease [15,21]. Likewise, purified P_aib_ could be beneficial in agricultural applications, such as biocontrol in post-harvest products, where it is better to apply them as a microbial-free treatment to avoid contamination of the final product. 

The present work aimed to produce P_aib_ for their extraction and characterization as a potential biofungicide. The work included the optimization of fungal growth conditions for P_aib_ production. Afterward, mass-spectrometry techniques were applied for the identification and sequencing of P_aib_. In addition, the biological effect of the biofungicide was evaluated against four phytopathogenic fungi in vitro and in vivo in tomatoes infected with *Alternaria alternata*. Furthermore, electron microscope images were used to study the effect of P_aib_ on the structure and morphology of the treated fungi. 

## 2. Materials and Methods

### 2.1. Fungi

The fungi *F. oxysporum*, *C. gloeosporioides*, *A. alternata*, * T. asperellum* and *B. cinerea* were obtained from the Costa Rican National Institute of Agricultural Technology Innovation and Transfer (INTA). *Trichoderma asperellum* was isolated from agricultural soil and used for P_aib_ production. *T. asperellum* was identified using ITS, rpb2 and tef1 sequences as previously described by Cai et al. [37]. These organisms were stored in ultrapure water at 4 °C.

### 2.2. Optimization of the Fermentation Media for P_aib_ Production

#### 2.2.1. Inoculum Preparation 

*T. asperellum* was seeded in potato-dextrose-agar (PDA) (Difco^TM^ Laboratories, Detroit, MI, USA) and incubated at 28 °C for one week. Then, a filtered spore suspension (1 × 10^6^ spores mL^−1^) was prepared as inoculum by flow cytometry. 

#### 2.2.2. Fermentation Media and Carbon Source Test

The growth media consisted of a carbon source (either glucose or sucrose at 30 g L^−1^), KNO_3_ (0.7 g L^−1^), NaNO_3_ (1.4 g L^−1^), MgSO_4_ · 7H_2_O (1 g L^−1^), KH_2_PO_4_ (0.8 g L^−1^), FeSO_4_ · 7H_2_O (0.01 g L^−1^), MnSO_4_ · H_2_O (0.01 g L^−1^) and CuSO_4_ (0.005 g L^−1^). Sterile flasks with 200 mL of medium were inoculated with a spore suspension of *T. asperellum* and incubated at 200 rpm during 21 days at 28 °C. Three replicates were prepared for each treatment. Every two days, samples were taken for measuring biomass production (dry weight), sugar consumption by HPLC and the production of peptides by mass spectrometry. After the evaluation, analysis of variance (ANOVA) and Tukey analyses were performed to select a carbon source for further experiments.

#### 2.2.3. Elicitor Addition Test 

The phytopathogenic fungi *C. gloeosporioides*, *F. oxysporum* and *B. cinerea* were grown in potato-dextro-broth (PDB) at 200 rpm during 7 days at 21 °C. The autoclaved and lyophilized cell debris of the three fungi were evaluated as elicitors of P_aib_ production. Each one was added to the fermentation media on the first day (1 g L^−1^). The fermentation conditions were maintained. Every two days, samples were taken and analyzed by the same methods. A control without cell debris was also prepared. Tests were carried out in triplicate for each fungus. Results were evaluated by using ANOVA and Tukey tests.

#### 2.2.4. Amino Acid Addition Test

Seven amino acids leucine (leu), proline (pro), valine (val), glycine (gly), alanine (ala), 2-aminoisobutyric acid (Aib) and glutamine (glu) were tested to evaluate their effect on P_aib_ production. Each amino acid was added separately to a flask on day 9 of the fermentation (1 g L^−1^). Tests for each amino acid were carried out in triplicate. Every two days, samples were taken and analyzed by the same methods. A control without amino acids was prepared in triplicate. Results were evaluated by using ANOVA and Tukey tests.

### 2.3. Fermentation Process Modeling

#### 2.3.1. Model Approach

A Central Composite Design (CCD) was applied to evaluate the statistical effects of the concentration of Aib and the concentration of the elicitor *F. oxysporum* against the P_aib_ production as the response. The axial values were codified as −α and +α which represent the lower and higher values for each factor (Table 1). The factorial values were codified as −1 and +1 and calculated by Equation (1), where ${Χ}_{i}$ is the value (unitless) of the variable, $\chi_{1}$ the real value of the variable, *X*_0_ the real value in the central point and *k* the number of independent factors:(1)$${Χ}_{i}=\frac{\left|\chi_{1}-\chi_{0}\right|}{{\left(2^{\kappa }\right)}^{\frac{1}{4}}}\;\;\;\;\;\;\;\;\;$$

A CCD factorial 2^2^ was applied including four factorial points, four axial points and five repetitions of the center point for a total of 13 runs. The fermentation was performed using the same conditions and induction days as above. A second order polynomial Equation (2) was used to calculate the predicted response:(2)$$\Upsilon =\beta_{0}+\sum_{i=1}^{\kappa }\beta_{i}\chi_{i}+\sum_{i=1}^{\kappa }\beta_{ii}\chi_{i}^{2}+\sum_{i<j}^{\kappa }\beta_{ij}\chi_{i}\chi_{j}+\varepsilon \;.$$
where Υ is the predicted response; $\chi_{i}$ and $\chi_{j}$ the input variables; *β_i_* the linear effects, *β*_0_ the intercept; *β_ii_* the quadratic effects; *β_ij_* the interaction; and ε the error.

The regression and graphical analysis were performed using Design Expert 12 of Stat-Ease. The optimal level of combinations was obtained after resolving the equation and analyzing the response surface graph by the Contour Profiler tool of the software. The lack of fit (*p* > 0.05), *R*^2^ > 0.9 and model significance (*p* < 0.05) were used to determine the goodness of fit.

#### 2.3.2. Model Validation

To validate the model, ten points were determined and used for a second experimental trial. These points consisted of the eight central points with respect to the edges and vertices of the graph plus two repetitions of the maximum point for P_aib_ production. The validation indexes’ accuracy factor ($A_{f}$) and bias factor ($B_{f}$) indicate the relation between the predicted and experimental data. The indexes were calculated according to Baranyi et al. [38], using Equations (3) and (4):(3)$$A_{f}=10\left[\sum \left|log\left(\frac{predicted}{experimental}\right)\right|/n\right].$$
(4)$$B_{f}=10\left[\sum log\left(\frac{predicted}{experimental}\right)/n\right]$$

### 2.4. Mass Spectrometry

To purify the P_aib_ produced, fermentation samples were centrifuged at 3000rpm for 10 min and filtered (0.45 µm). The filtrate was loaded into Visiprep™ SPE Vacuum Manifold with C18 cartridges (Supelco Analytical Empore™ SPE) [39]. Contaminants were removed by 4 volumes of osmosis water. The P_aib_ were eluted using ethanol (96% *v*/*v*^−1^) [40]. The ethanol was removed using a vacuum concentrator (SpeedVac). The dried samples were dissolved in solution 1 (HPLC grade methanol 75% *v*/*v*^−1^, osmosis water 24.9% *v*/*v*^−1^, and formic acid 0.1% *v*/*v*^−1^) and filtered (0.2 µm nylon).

The samples were analyzed on a mass spectrometer (MDS SCIEX Applied Biosystems 4000 Qtrap HPLC MS/MS, Waltham, MA, USA) to determine the proportions of the metabolite in each one. The mobile phase corresponded to a mixture of MilliQ water and HPLC grade methanol, both with formic acid (0.1% *v*/*v*^−1^) to support the protonation of the ions. The HPLC conditions were as follows: Agilent^®^ 1200 (Santa Clara, CA, USA), detector: mass spectrometer, column: XDB Agilent^®^ C18, 50 mm × 4.6 mm; 1.8 μm, oven temperature: 25 °C, column temperature: 25 °C, flow: 450 μL min^−1^ and injection volume: 28 μL. HPLC gradient was used for sample analysis (Table 2). 

The search for masses was carried out by quadrupole 1, in a mass interval of 200 to 2000 *m*/*z*, in positive mode, for which the following parameters were used: Curtain gas (CUR): 26 psi; Internal standard (IS): 5500 IS; Source temperature (TEM): 250 °C; Ion source gas 1 (GS1): 23 psi; Ion source gas 2 (GS2): 19 psi; Ihe: on and collisionally activated dissociation (CAD): medium.

A MS analysis was performed to determine the retention times and the area of the P_aib_ peak, as well as the abundance of these in the samples. The mass spectrum was analyzed using Analyst^®^ software version 1.6.2. The best treatment consisted of the one that produced the highest intensity of the P_aib_ of interest: trichotoxins.

### 2.5. P_aib_ Sequencing

The P_aib_ samples obtained from the experiments in Section 2.3 were purified using the methodology in Section 2.4. The sample was injected directly into the electrospray source using a Hamilton syringe [41]. The MS/MS spectrum for each P_aib_ was obtained by analyzing the mass spectra and precursor ions’ fragmentation.

For this purpose, the operating parameters of both pieces of equipment were used, as previously established in Section 2.4. Once the spectra for each P_aib_ was obtained, they were sequenced manually based on previously reported sequences.

### 2.6. Antifungal Activity of P_aib_ from T. asperellum

#### 2.6.1. Extract

A P_aib_ extract was obtained from a fermentation running at the optimal conditions determined in Section 2.2. After harvesting, the broth was vacuum filtered (Whatman 1) and purified four times by a liquid–liquid extraction system with ethyl acetate (3:1 *v*/*v*). The ethyl acetate phase was recovered. The solvent was eliminated by rotatory evaporation and the P_aib_ extract was lyophilized. A biofungicide prototype was formulated with the produced P_aib_.

The components of the extract included ethanol 96% (44% *v*/*v*^−1^), citrate buffer (44% *v*/*v*^−1^, pH 5.6), Tween 20% (12% *v*/*v*^−1^) and P_aib_ extract (139,400 µg mL^−1^). Additionally, a control extract was prepared without the P_aib_ extract. 

#### 2.6.2. Pathogenic Fungi In Vitro Growth Inhibition 

Growth inhibition tests were performed to confirm the antifungal activity of the extract on four phytopathogenic fungi. For this, three different treatments were developed in triplicate for each fungus: (1) PDA with the extract (800 µg ml^−1^); (2) PDA with the control extract (800 µg ml^−1^) and (3) PDA with clotrimazole (800 µg ml^−1^) as a positive control. The P_aib_ concentration 800 µg ml^−1^ was previously identified in our laboratory (not shown) as the MIC (Minimum Inhibitory Concentration) for the evaluated fungi.

The fungi were grown by placing a mycelial disc (1 cm) in the center of the Petri dish and incubated at 28 °C. The radial growth of the fungi was measured to obtain the percentages of growth inhibition using Equation (5), where GH: growth inhibition (%), *C*: control growth (cm) and *T*: treatment growth (cm). The test was stopped once the fungi reached the edge of the Petri dish.
(5)$$\mathrm{GH}\;\left(\%\right)=\frac{C-T}{C}\times 00\;.$$

Statistics and graphics were performed using R Core Team (2020). The inhibition effect of each treatment was analyzed using a one-way ANOVA.

#### 2.6.3. *A. alternata* Growth Inhibition in Tomatoes 

The surface of the tomatoes (*Solanum lycopersicum*) was sterilized by washing it with sterile, distilled water and soap. Then, the tomatoes were sprayed with 70% ethanol and left to dry for one hour in a laminar flow cabinet. After that, four 1.5 cm diameter cross-shaped wounds were made with a sterile needle around the top of the tomato. Subsequently, the tomatoes were inoculated by injecting a suspension of 1 × 10^6^ spores mL^−1^ of *A. alternata* on each wound. A time of 30 min was given in a laminar flow cabinet for the wound to absorb the suspension. 

The inhibitory effect of P_aib_ on the growth of *A. alternata*, on the infected tomatoes was evaluated using four different treatments: (1) solution of the extract with P_aib_ (2 mg mL^−1^); (2) solution of the control extract (2 mg mL^−1^); (3) sterile distilled water and (4) a solution of Clotrimazole (2 mg mL^−1^). The solutions were prepared by dissolving the required quantities of each treatment into sterile, distilled water. The treatment solutions were injected into the tomatoes’ wounds, left to rest for 30 min in a laminar flow cabinet and placed in separate sterile boxes according to their treatment at 23 ± 2 °C. Ten tomatoes were used for each treatment. The growth inhibition was measured with the incidence of the disease, i.e., the number of wounds infected, and the diameter of the lesion of infected wounds. Data collection was completed on day eight after infection. 

### 2.7. Effect of P_aib_ on the Morphology of Phytopathogenic Fungi 

#### 2.7.1. Sample Preparation

Scanning electron microscopy (SEM) images were obtained to observe the effect of P_aib_ on the morphology and structure of 4 phytopathogenic fungi. The microorganisms were cultured on PDA plates supplemented with 800 µg mL^−1^ of P_aib_ extract or 800 µg mL^−1^ of control extract and incubated for 8 days at 28 °C. Subsequently, a sample was taken from each plate by extracting the mycelium with a needle and placed in a 5 mL glass vial for processing.

#### 2.7.2. Sample Fixation 

The vials containing the samples were fixed with a solution composed of 2% glutaraldehyde, 2% formaldehyde and phosphate buffer (PB) 0.1 M pH 7.4 and stored for 4 h at 4 °C. Afterwards, 2 mL of a PB 0.05 M were added to each sample and the samples were placed for 10 min in an orbital shaker (80 rpm). Following that, the vials were decanted, and the supernatant was discarded. The washing of the mycelium was repeated two times more. Then, 2.3 mL of OsO_4_ at 2% in PB 0.05 M were added to the vials and these were placed in an orbital shaker (80 rpm) for 16 h. Finally, the supernatant was discarded, and 3 washes were made with PB 0.05 M as previously indicated.

The procedure consisted of adding 2 mL of ethanol at different percentages (30%, 50%, 70%, 80%, 90%, 95% and twice at 100%) and letting it stand for 15 min each, except at 100% which rested for 20 min. Excess alcohol was removed from the sample with a pipette. Subsequently, the samples were dispensed into 1.5 mL Eppendorf tubes and dried in an oven at 40 °C for 4 days. The samples were placed on aluminum bases with carbon-aluminum tape. Then, the samples were covered with gold (AU) on the DENTON VACUUN DESK V (Moorestown, NJ, USA) ionic blanket at 30 mA/180 secs (EMS 550X Sputter Coater: 50 mA 2:30 min 1 × 10^−1^ mbar). Finally, samples were observed by SEM JEOL JSM-6390 LV, Tokyo, Japan (Voltage acceleration: 10 KV, Secondary electrons: SEI and Spot Size: 50).

## 3. Results and Discussion

### 3.1. Optimization of Fermentation Media for P_aib_ Production 

#### 3.1.1. Carbon Source Utilization Test

The effect of glucose and sucrose on P_aib_ production was evaluated. The variation in the biomass and P_aib_ production is shown in Figure 1. The addition of sucrose to the culture medium significantly increased the production of P_aib_ (*p* = 0.003) while biomass generation was reduced. Conversely, the addition of glucose to the culture medium caused an increase in growth but lowered P_aib_ production.

Sucrose is a disaccharide composed of a glucose molecule plus a fructose molecule, so it requires hydrolysis by an invertase before glucose enters the glycolysis pathway [42]. Most likely, the fungal growth was lesser than in the sucrose sample because the amount of glucose, available for primary metabolism, was limited to half compared to the glucose medium. On the other hand, glucose as a carbon source is preferred by most microorganisms as it does not require other catabolic processes to enter the glycolysis pathway [43]. 

High extracellular glucose concentrations act as a signal to the cell that external conditions are favorable for cell growth and reproduction, characteristic of the exponential phase of growth. However, this signaling represses the expression of some genes related to the secondary metabolism of the microorganism involved in its survival under unfavorable conditions, such as those that occur during the stationary phase [42]. The decrease in P_aib_ productivity could be explained by the presence of glucose sensor homologs and transcriptional regulators that negatively regulate genes encoding for NRPS when saturated with glucose.

This hypothesis is supported by the study of Zhou et al. [44], which determined that in *Trichoderma longibrachiatum* SMF2, the transcriptional regulator TlSTP1 is responsible for the negative regulation of genes encoding for NRPS and the positive regulation of hexose transporters. This regulator possesses a conserved glucose transporter domain with apparent function as a sensor of this monosaccharide. Deletion of the gene coding for this protein caused a decrease in the vegetative growth of the fungus related to a deficiency in glucose capture by a change in the expression of 20 glucose transporters. However, P_aib_ production increased and started two days earlier, which is related to the increased expression of NRPS encoded by the *tlx1* and *tlx2* genes. Phylogenetic analyses have demonstrated the presence of TlSTP1 homologs in P_aib_-producing species such as *T. asperellum* with a high sequence identity of 87–96% [44].

Sucrose assays evidenced a significant increase in P_aib_ production over glucose. Increased production of P_aib_ could be due to the low saturation of glucose sensors and therefore, the reduction in the negative regulation of NRPS genes. From a commercial point of view, sucrose is more advantageous because of its high availability and low cost. In addition, purification processes are facilitated and are more efficient by having less biomass as a by-product of the bioprocess. Thus, the use of sucrose as a carbon source is considered a better option for productivity, scale-up and cost reduction of this fermentation.

#### 3.1.2. Elicitor Addition Test

The use of fungal debris as an elicitor is based on the ability of *Trichoderma* to recognize the presence of other surrounding microorganisms. The constant release of lytic enzymes allows the sensing of molecules, such as oligopeptides and oligochitosaccharides, from the cell membrane of other fungi. This identification activates the regulatory transcription factors related to the release of bioactive secondary metabolites, as well as mycoparasitism that exploits the host as a source of nutrients [43]. Thus, simulating the presence of another microorganism in the culture medium can stimulate the activation of pathways related to antibiosis and mycoparasitism [45].

The cellular debris of phytopathogenic fungi, which showed sensitivity against P_aib_ from *T. asperellum*, were used as elicitors [46,47]. By day nine of fermentation, all of the treatments evidenced a significant increase in P_aib_ production (*F* = 28.11, *p* < 0.001) compared to the control, as shown in Figure 2.

Tamandegani et al. [48] found that the direct in vitro interaction of *T. asperellum* with other plant pathogens enhanced P_aib_ productivity. Furthermore, Tamandegani et al. [48], determined that P_aib_ production increased significantly upon in vitro interaction with *F. oxysporum*. In this study, greater production of the peptides was obtained when *F. oxysporum* cell debris was added to the culture medium (Figure 2). *Botrytis* cell debris also contributed to a significant increase in P_aib_ production compared to the control; however, the production peak had a lower intensity and occurred five days after the peak caused by *F. oxysporum* treatment. 

The presence of elicitors also influenced sucrose consumption, which was higher in all of the treatments against the control (Figure 2). *T. asperellum* interprets cellular debris as the presence of another fungus in the culture medium, triggering a competitive growth mechanism and thus a quicker consumption of carbon preventing the growth of the phytopathogen [49]. Sucrose concentration by day nine in the *F. oxysporum* treatment was reduced to 2.54 g/L, indicating that most of this sugar had been consumed and the fungus had reached stationary phase where it produces secondary metabolites such as P_aib_ [50,51]. The addition of *F. oxysporum* to the culture medium as elicitor was selected to increase P_aib_ production.

#### 3.1.3. Amino Acid Addition Test

A group of amino acids was selected based on the frequency of their presence in the structure of P_aib_ produced by the genus *Trichoderma* (P_aib_ Database [52]). These were added to the culture medium on day 9 of fermentation because most of the sucrose in the medium was consumed and the fungus entered the stationary phase on this day (Figure 3). Moreover, the addition of the amino acids at the beginning of the stationary phase prevented them from being directed to other pathways and reactions inherent to the primary metabolism. This procedure ensured the availability of amino acids to be incorporated into the structure of P_aib_ [51]. 

The independent addition of the amino acids Aib, Val and Pro significantly increased the production of P_aib_ on day 21 (*F* = 6.22, *p* < 0.001), as shown in Figure 3. However, only Aib showed a significant difference in P_aib_ intensity in relation to the control. 

The addition of Aib increased P_aib_ production due to its immediate availability in the culture medium [53]. This amino acid is the main component within P_aib_ of *Trichoderma* with a relative abundance close to 37% (Appendix A). When Aib is already available, the fungus does not have to synthesize it which facilitates the formation of peptide chains.

The biogenesis of Aib is based on a methyltransferase reaction using adenosyl methionine as a methyl group donor to an L-alanine molecule [19]. L-alanine is primarily used in protein biosynthesis, so its availability for Aib biosynthesis is limited. Despite being the precursor amino acid of Aib, L-alanine did not significantly increase P_aib_ production (Figure 3). The amino acid Aib increased the synthesis of P_aib_ of *T. asperellum* when added to the culture medium during the stationary phase.

### 3.2. Fermentation Process Modeling 

The concentration of Aib and the elicitor *F. oxysporum* were selected as the factors to be evaluated in the modeling process for the optimization of P_aib_ production. The sucrose concentration (30 g/L) was maintained as a fixed condition in the fermentation. The central composite design with a 2^2^ factorial distinguished the specific concentrations to be evaluated, as shown in Table 3, in a range of 0.5 to 3 g/L for both independent variables. A value lower than 5 g/L Aib was used, as fungistatic activity against other fungi has been reported at this concentration [54].

The model showed a coefficient of determination (*R*^2^) of 0.9245 and a *p* = 0.008 (*p* < 0.05), which indicates that 92.45% of the total difference in the response is explained by this model. A *R*^2^ value close to 1.0 indicates that there is little difference between the experimental values and the predicted values, so the model is considered significant. The lack of fit obtained was not significant with *p* = 0.6950. These data suggest that the second-degree equation obtained explains the production of P_aib_ under the conditions evaluated:(6)$$\;y\;=4.28\times 10^{8}\;A_{1}+1.45\times F_{2}^{2}+4.05\times 10^{8}\;.$$

It was determined that the linear factor of Aib concentration (*A*) and the quadratic factor of *F. oxysporum* concentration (F-F) exert a significant effect (*p* < 0.05) on the dependent variable (Appendix A). It should be noted that the model did not identify synergistic or antagonistic interactions between the factors evaluated. The second order polynomial equation obtained is shown in Equation (6), where y is the P_aib_ production response, *A* is the concentration of Aib and F is the concentration of *F. oxysporum*.

Figure 4 shows the surface response graph obtained with the central composite model where the maximum point of P_aib_ production is located at 2.634 g/L Aib and 0.866 g/L *F. oxysporum*.

The validation of the model was carried out using the concentrations of Aib and *F. oxysporum* shown in Table 4. The validation of the model resulted in a certainty level of 1.288 and a bias factor of 0.997, which indicates that the experimental values correlate well with the predictions provided by the model.

In another paper, a response surface model was developed to predict the production of the P_aib_ Tricokonin VI from *T. koningii* SMF2 in solid-state fermentation. The factors of inoculum size, incubation temperature, humidity and initial pH were evaluated with the production of Trichokonins VI as a response variable. The equation obtained determined that all of the factors are representative and optimal values were defined for each of them with respect to P_aib_ production [40]. Both of the models affirm the possibility of optimizing the production of P_aib_ using central composite designs, which facilitates the scaling of these processes towards the industry.

### 3.3. P_aib_ Sequence and Identification 

The sample for sequencing was taken from the fermentation used to generate the model, which ensured that it contained a sufficient concentration of peptides to perform the analysis. Two groups of P_aib_ were identified in *T. asperellum* including 38 asperelines and 5 trichotoxins [19,34,55], however only the last ones were obtained. Each trichotoxin was manually sequenced based on the fragment ions generated and the sequences reported in the literature. The sequencing of the P_aib_ was carried out by positive mode ion cleavage using ESI-MS/MS. 

The mass spectra showed the presence of ions characteristic of trichotoxins with values between 1676 *m*/*z* and 1768 *m*/*z*. The fragments detected corresponded to ions with *m*/*z* 1676, 1691, 1704, 1705, 1718, 1726, 1742 and 1768. The fragmentation patterns of the trichotoxins were obtained, except for the ion 1742 and 1768 *m*/*z*.

As trichotoxins (SF1 P_aib_ subfamily) are synthesized by a 18-module NRPS [56], they are composed of eighteen amino acids [48]. Previously sequenced trichotoxins have shown the general sequence: Ac-Aib-Gly-Aib-Lxx-Aib-Gln-Aib-Aib-Aib-Aib/Ala-Ala-Aib/Ala-Aib-Pro-Lxx-Aib-Aib-Aib/Vxx-Gln/Glu-Valol. The reduced specificity and three-dimensional structure of NRPS yields large number of homologous and isomeric P_aib_ [32]. This group of peptides possesses 4 microheterogeneities at positions 9, 11, 16 and 17, resulting in the production of at least 5 distinct trichotoxins (Table 5).

The ESI-MS/MS mass spectrometry method does not allow for the establishment of a difference between isobaric amino acids such as leucine and isoleucine, and valine and isovaline, so they are shown as Lxx and Vxx, respectively [56].

The relative amino acid composition may vary depending on the availability of precursors and free amino acids, which could favor the production of one trichotoxin over the other [53]. The addition of Aib to the culture medium favored the synthesis of the trichotoxins A-40 and A-50G, while all of the other amino acids and the control resulted in a higher production of trichotoxins T5D2 and 1703A. Trichotoxin 1705 has two more Aib residues than trichotoxins T5D2 and 1703A (Table 5), so the addition of Aib to the medium may favor the production of this trichotoxin due to increased availability. 

The microheterogeneity given by the flexibility of some NRPS modules allows for the obtaining of many trichotoxin isoforms in *T. asperellum*. These isoforms can vary from each other by a single mass unit as occurs between trichotoxins 1703A and A-40 [53]. In this case, the difference occurs by residue substitutions at positions 11 and 16, where trichotoxin 1703A has Ala and Vxx residues, respectively, while trichotoxin A-40 III has Aib residues in both positions. Microheterogeneities among the trichotoxins identified were at positions 9, 11, 16 and 17 with Ala/Aib, Ala/Aib, Vxx/Aib and Glu/Gln variations, respectively. Glutamine residues in the sequence are related to the formation and stabilization of the ion channel, whereas glutamate residues may have an impact on its destabilization. Therefore, trichotoxins with two glutamine residues, such as trichotoxin 1717A, have higher biological activity [59].

Different *T. asperellum* strains have been reported to produce P_aib_. An aspereline-producing marine strain with a Prolinol residue at its C-terminus has been reported [55]. A terrestrial strain TR356 produces the same asperelines and some trichotoxins [57]. On the other hand, the strain used in this assay appears not to produce asperelines but releases some trichotoxins different from strain TR356. The *T. asperellum* strain used by Sood et al. [53] produced trichotoxins 1717A and 1703A which were also produced in this assay, but they did not report the production of other trichotoxins. 

These differences may be due both to varied growing conditions as well as the purification and identification techniques used in the different assays by the research groups. In addition, intraspecific differences can vary the production of these peptides, depending on the environment in which each fungus develops.

### 3.4. Antifungal Activity of P_aib_ from T. asperellum

#### 3.4.1. Pathogenic Fungi In Vitro Growth Inhibition 

The biological activity of the P_aib_ extract was evaluated in vitro against *C. gloeosporioides, B. cinerea, A. alternata* and *F. oxysporum*. These fungi were selected due to their negative effects in agriculture, related to diseases in crops for human consumption that cause economic and health damage [60,61,62]. Previous studies from our laboratory (data not shown) determined that the P_aib_ MIC for these fungi at 800 µg mL^−1.^ The extract showed an evident effect against the growth of all the pathogenic fungi tested (Figure 5). According to the GH obtained at the end of each test, the extract demonstrated to be more effective against *C. gloeosporioides* with a GH of 92.2%. The GH for *B. cinerea, A. alternata* and *F. oxysporum* were 74.29, 58.4 and 36.2%, respectively. 

In relation to the clotrimazole, as control treatment no significant differences (*p* > 0.05) were observed against *C. gloeosporioides* and *A. alternata* when applying the P_aib_ extract. Clotrimazole, at 800 µg mL^−1^, completely inhibited the growth of *B. cinerea* and *F. oxysporum* whereas some growth was still observed in the plates treated with the P_aib_ extract. 

The differences between the clotrimazole and the P_aib_ extract effectiveness are determined by their mode of action and the specific fungi [63,64]. Clotrimazole inhibits the biosynthesis of ergosterol, a key membrane component, by altering the permeability of the fungal cell wall [65]. In addition, clotrimazole inhibits the enzyme Ca^2+^-ATPase of the sarcoplasmic reticulum, depletes the intracellular calcium and blocks the calcium-dependent potassium channels [66]. 

The mechanism of action of P_aib_ consists in the formation of pores or voltage-dependent ion channels in bilayer lipid membranes. These transmembrane channels conduct the inward current of ions and, together with the structural changes in the surface of the membrane, allow an uncontrolled exchange of cytoplasmatic material causing cell death [39,63,67,68,69]. Additionally, some P_aib_ inhibit the activity of the B-Glucan synthase, an essential enzyme in the formation of the fungal cell wall [70,71,72]. 

The differences observed for the inhibition of the P_aib_ extract against each phytopathogenic fungus were somehow expected, since their effect on the cell membrane can vary amongst microorganisms [39,63,73,74]. This is determined by the elasticity, structure, lipid composition and charge of the fungal membrane as well as the peptide/lipid (P/L) molar ratio [30,63,64].

Likewise, the ion channel structure may vary according to the P_aib_ that form them, for example, the ion channels formed by the trichotoxin_A50E have a different shape and conductance properties than the ones formed by alamethicin [72]. Long-chain P_aib_ form voltage-dependent and non-voltage-dependent ion channels, while short-chain P_aib_ are not long enough to insert into the membrane and instead form aggregates that destabilize the membrane [75].

This difference in the degree of effect/inhibition of P_aib_ according to structure is reinforced by comparing the data of Grigoletto et al. [76] on inhibition in *C. gloeosporioides* with P_aib_ of the trilongins BI-BIV group that produced an inhibition of 44.97% or less. The P_aib_ isolated and identified in this work correspond to the group of trichotoxins, in this case, the percentage of inhibition was also remarkably high (92.2%). This indicates that trichotoxins are an appropriate type of P_aib_ to fight against infections by *C. gloeosporioides*.

#### 3.4.2. *A. alternata* Growth Inhibition in Tomatoes 

The antifungal activity was tested in tomatoes inoculated with *A. alternata*. This fungus was selected because it is one of the pathogens that commonly attacks several crops intended for human consumption, such as tomato, potato and citrus [77,78,79]. In addition, *Alternaria* spp. are of agricultural importance because they cause worldwide economic losses and they are difficult to control despite repeated and intensive application of fungicides [5,80,81,82,83]. 

The growth of *A. alternata* was observed from day 2 in the tomatoes treated with the control and in the samples that were injected with water. Eight days after inoculation, the incidence of infection for the samples treated with water was 100 %. The lesion diameters of the infected tomatoes were measured in the treatment of the extract without P_aib_ and of the samples treated with water; the average was 1.73 ± 0.62 cm and 1.62 ± 0.55 cm, respectively. These data do not show significant differences between each other (t = 0.83; gl = 75; *p*-value = 0.21). In contrast, the tomatoes treated with the extract showed no growth of the fungus, even after 15 days, which suggests that the extract is not only effective in the short term but that it can show a continuous inhibitory effect even after two weeks. The incidence of the disease in tomatoes treated with the P_aib_ extract was 0% (the same as with clotrimazole), while the untreated fruit (extract without P_aib_) showed a 92.5% incidence of infection. 

Figure 6 shows the state of the tomatoes after 8 days of infection, as observed, the spores did not germinate in the tomato treated with P_aib_. Thus, the P_aib_ extract not only inhibits mycelial growth but also hinders spore germination which broadens the applications of P_aib_ as a biofungicide, since in nature, diseases are commonly transmitted from plant to plant by spore dissemination [65,84]. It has been previously reported that the P_aib_, trichohorzianine A1, can inhibit the germination of *B. cinerea* spores at 30 h [85]. In this work, the inhibitory effect on spore germination was maintained even after 15 days, which suggests that the spores lost their viability. This result is very promising since *Alternaria* spp. spores are very resilient and have even shown resistance to certain fungicides [83]. 

These results suggest that the P_aib_ extract could be used as a biofungicide to treat *Alternaria* spp. pests, since it inhibited both mycelial growth and spore germination. Moreover, the application of extracts containing P_aib_, (unlike using *Trichoderma* spp. as biocontrollers), could avoid the activity against non-targeted species, which eventually gives it an advantage as a biofungicide [86]. Another benefit of using P_aib_ as a biofungicide is that exogenous application of P_aib_ can induce multiple metabolic activities that allow the plant to increase resistance against pathogens by activating the plant’s defense responses [87,88,89]. 

It must be considered that the extract could contain other minor components that contribute to antifungal activity. However, *T. asperellum* is a well-documented antifungal P_aib_ producer [55]. These peptides were identified in the P_aib_ extract, assuring these peptides are present in determined concentrations. The results suggest the antifungal activity of the extract is mainly produced by P_aib_. 

#### 3.4.3. Effect of P_aib_ on the Morphology of Phytopathogenic Fungi 

SEM images were taken to observe the effect of P_aib_ on the morphology and structure of the phytopathogenic fungi. The images showed noticeable differences between the structures of the fungi treated with P_aib_ and the control (Figure 7). While the control showed hyphae with smooth surfaces and normal conidia, the images of the treated fungi demonstrated the clear effect of the P_aib_. 

The images of all of the samples treated with the extract showed dehydrated hyphae with granules and an evident damage to the fungus wall. The action mechanism of P_aib_ consists of the formation of permanent transmembrane pores due to their amphipathic nature; this causes the escape of cytoplasmic material and eventual cell death [11,14,32]. The endoplasmic material that exits through the pores accumulates on the surface of the hyphae, generating granulated surfaces as observed in the hyphae treated with P_aib_. In addition, the outflow of material causes dehydration, which is why the treated hyphae appear wrinkled (Figure 7). 

Other reports have shown, using transmission electron microscopy, the effect of P_aib_ (trichokonin VI) on the cells of *F. oxysporum* where an accumulation of cytoplasmic vacuoles and swollen mitochondria with disrupted membranes was observed [90]. A similar behavior was reported in *F. oxysporum* when a culture filtrate of *Streptomyces griseorubens* was applied [91]. As in this work, SEM photos were used by Al-Askar et al. [91] to evaluate the effect of the antifungal compound on hyphal damage, with similar results as for the case of the P_aib_ extract. In addition, it has been reported that the hyphal elongation process can be affected by structural, non-reversible changes [91], which explains the inhibition of mycelial growth in the four treated fungi. 

The effect on conidia damage and deformation was also significant in all of the phytopathogenic fungi tested. This may explain why *A. alternata* conidia did not germinate in the tomato trials (Figure 7). It has been described that the viability of conidia can be affected by severe structural changes which may represent irreversible damage resulting in inhibition of germination [92]. The image of *C. gloeosporioides* (Figure 7(A2)), shows dehydrated and deformed conidia. 

## 4. Conclusions

The optimization of *T. asperellum* fermentation for P_aib_ production was carried out to determine the best carbon source, additive amino acid, elicitor and their optimal concentrations. As a result, sucrose consumption and P_aib_ production were significantly increased while biomass production and fermentation time reduced, which is beneficial for the scale-up and cost reduction of the bioprocess. P_aib_ were purified and identified as trichotoxins of 18 amino acid residues. The general sequence obtained corresponded to Ac-Aib-Gly-Aib-Lxx-Aib-Gln-Aib-Aib-Aib/Ala-Ala-Aib/Ala-Aib-Pro-Lxx-Aib-Aib/Vxx-Gln/Glu-Valol. 

The antifungal activity assays proved the efficiency of the P_aib_ extract to inhibit the growth of *C. gloeosporioides*, *B. cinerea*, *A. alternata* and *F. oxysporum* with a GH of 92.2%, 74.29, 58.4 and 36.2%, respectively. Additionally, the extract completely inhibited the germination of *A. alternata* spores on tomatoes. The SEM results showed how P_aib_ generates damage in the morphology of hyphae and spores of the treated fungi. These results indicate that the P_aib_ extract could be used as a growth inhibitor against phytopathogenic fungi of agricultural relevance. 

The results from this study suggest that environmental soil fungus from Costa Rica may represent an interesting source of known and new P_aib_ and antimicrobial compounds of biotechnological interest. Future studies may incorporate the determination of effective application levels in the field, the validation of proposed treatment against other species and strains and the best use strategy for the product.

## Figures and Tables

**Figure 1 jof-08-01037-f001:**
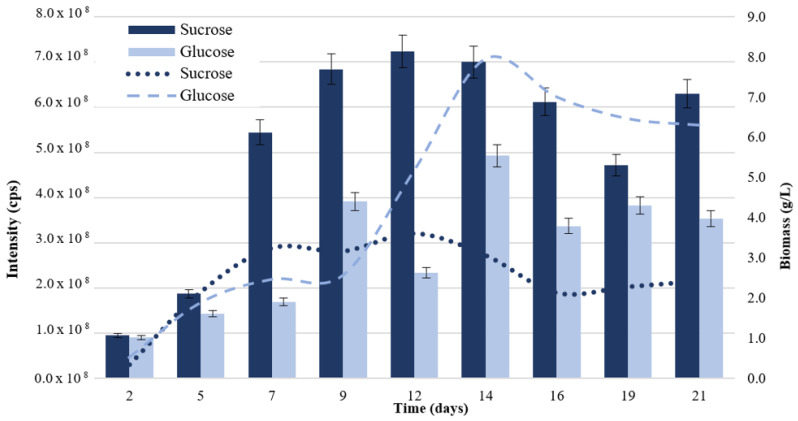
Production of P_aib_ (bars) and biomass (lines) of *T. asperellum* according to the carbon source added to the culture medium. Intensity values on the left axis are relative to P_aib_ production.

**Figure 2 jof-08-01037-f002:**
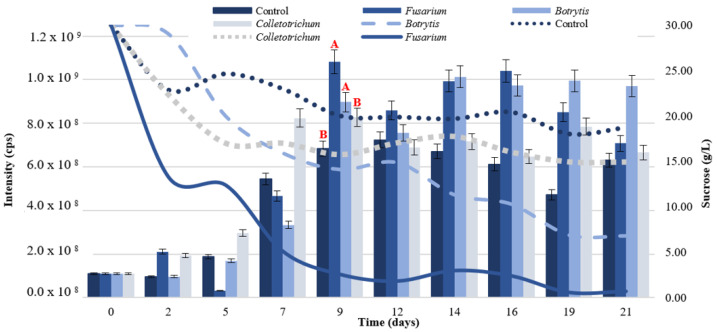
P_aib_ production (bars) and sucrose consumption (lines) of *T. asperellum* according to the elicitor added to the culture medium. Intensity values on the left axis are relative to P_aib_ production. Different letters represent significant differences between treatments.

**Figure 3 jof-08-01037-f003:**
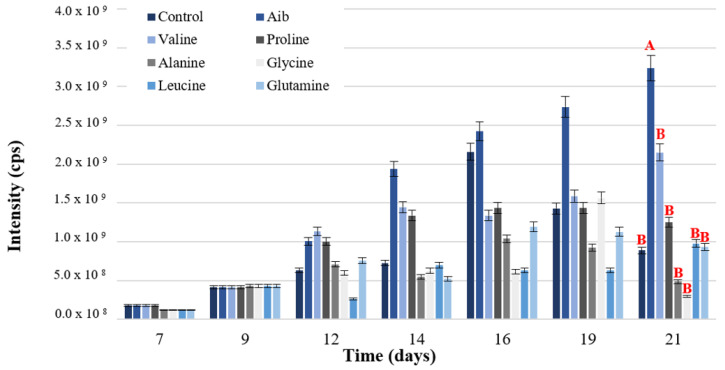
P_aib_ production of *T. asperellum* according to the amino acid added to the culture medium. Intensity values on the left axis are relative to P_aib_ production. Different letters represent significant differences between treatments.

**Figure 4 jof-08-01037-f004:**
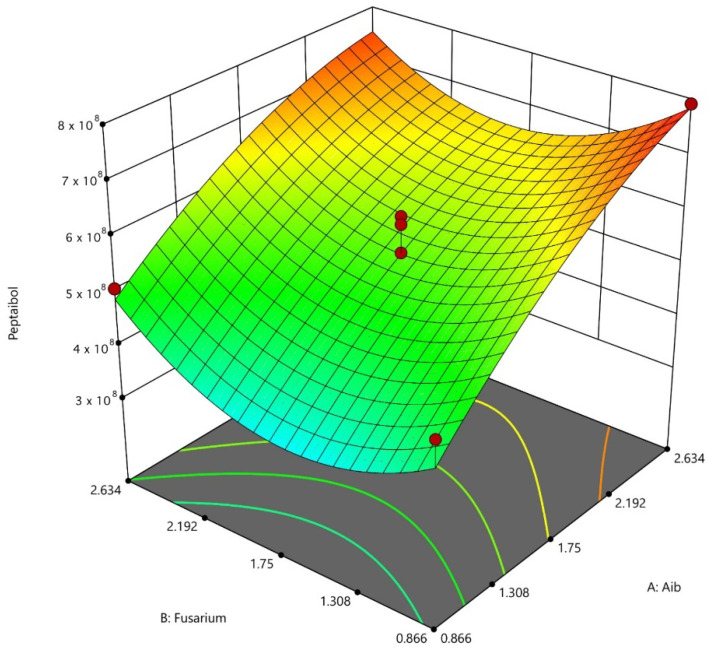
Surface response graph of the central composite model generated for the optimization of P_aib_ produced in the fermentation of *T. asperellum*.

**Figure 5 jof-08-01037-f005:**
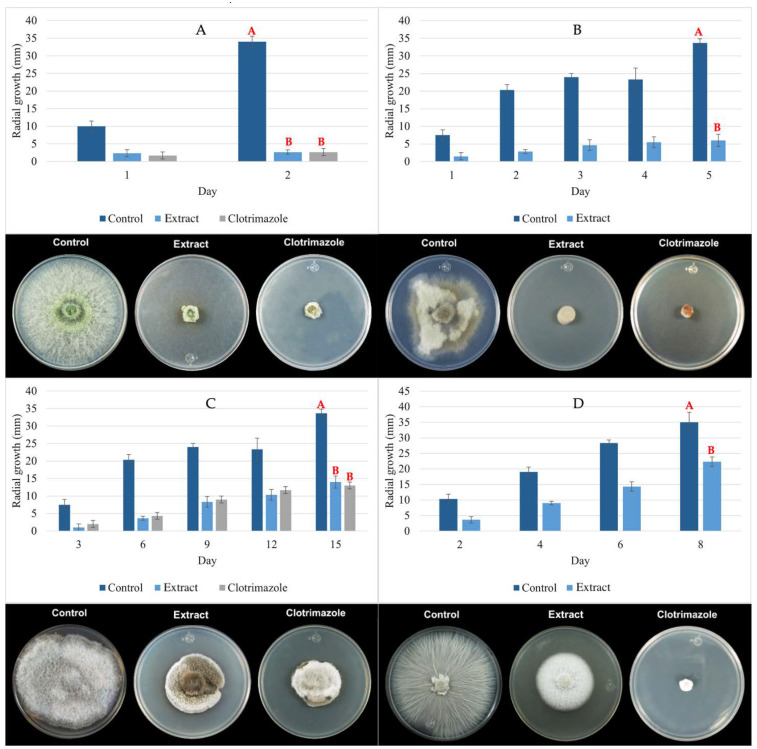
Inhibition effect of P_aib_ against mycelial growth of (**A**) *C. gloeosporioides;* (**B**) *B. cinerea;* (**C**) *A. alternata* and (**D**) *F. oxysporum* on PDA media after treatment with 800 µg mL^−1^ of P_aib_ extract. Different letters represent significant differences between treatments.

**Figure 6 jof-08-01037-f006:**
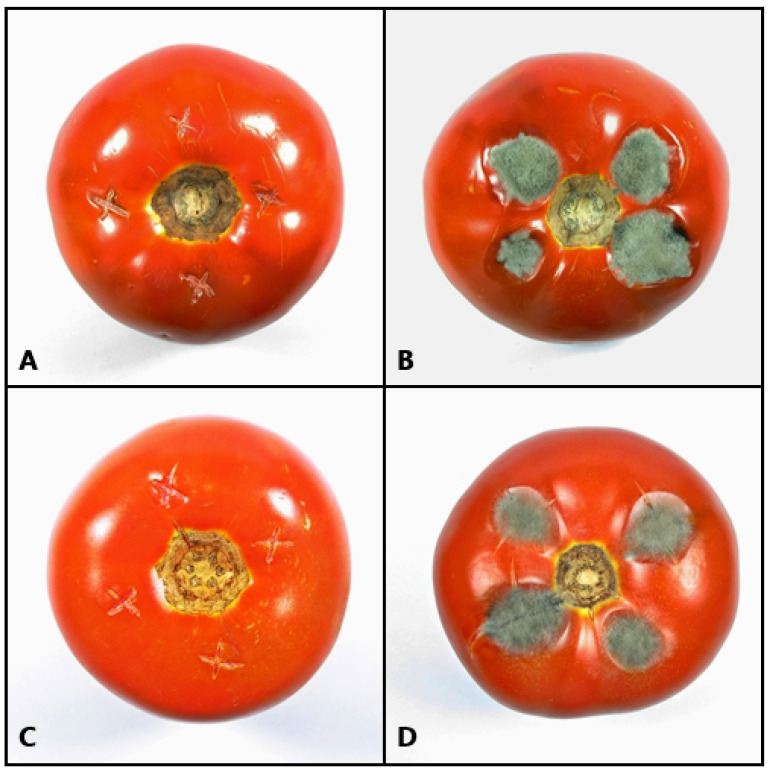
Effect of P_aib_ on growth of *A. alternata* in tomatoes 8 days after spore inoculation and application of treatment. (**A**) extract with P_aib_; (**B**) control; (**C**) clotrimazole and (**D**) sterile distilled water.

**Figure 7 jof-08-01037-f007:**
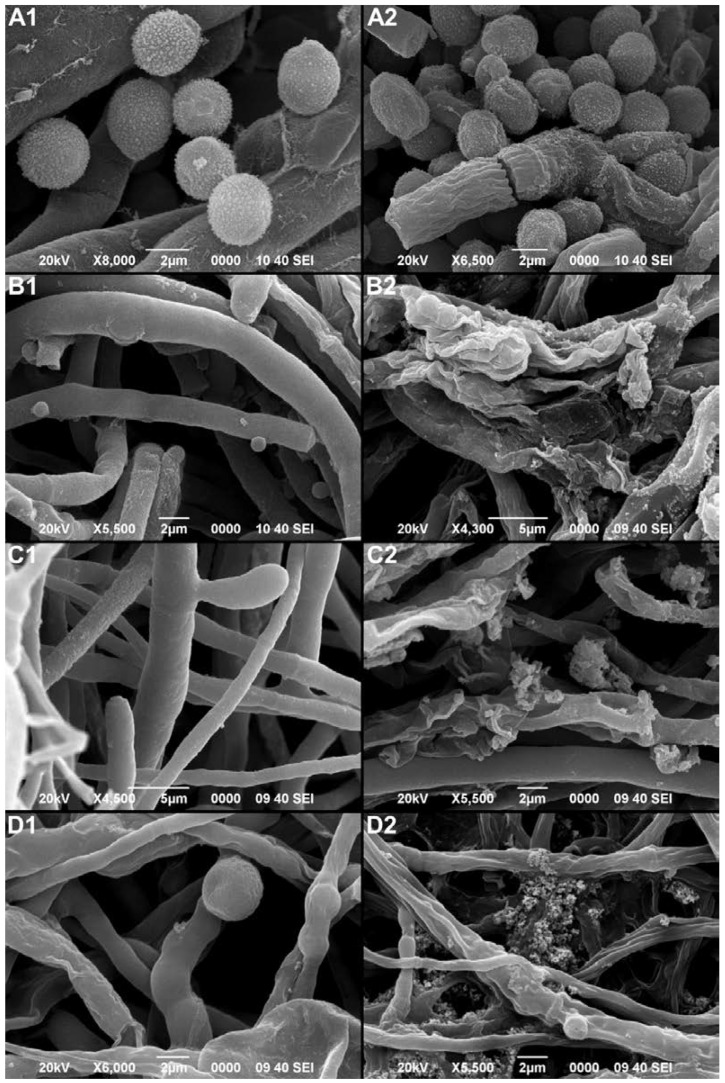
Scanning electron microscope images showing the effects of P_aib_ over the morphology of untreated fungus (**1**) and fungi treated with P_aib_ (**2**). (**A**) *C. gloeosporioides;* (**B**) *B. cinerea;* (**C**) *A. alternata* and (**D**) *F. oxysporum*.

**Table 1 jof-08-01037-t001:** Independent variable levels for the central composite design.

Variable	−α	−1	0	1	α
Aib	0.500	0.866	1.750	2.634	3.000
Fusarium	0.500	0.866	1.750	2.634	3.000

**Table 2 jof-08-01037-t002:** HPLC gradient for sample analysis.

Time (min)	A (%): Water/H+	B (%): Methanol/H+
0	30	70
16	15	85
25	0	100
35	0	100

**Table 3 jof-08-01037-t003:** Observed and predicted values for P_aib_ production for each experiment of the central composite model that evaluates the concentrations of Aib and *F. oxysporum*.

Trial	Treatment (g/L)		P_aib_ Production (cps)	
	Aib	*F. oxysporum*	Observed	Predicted
1	0.500	1.750	1.90 × 10^8^	2.34 × 10^8^
2	0.866	0.866	4.79 × 10^8^	4.29 × 10^8^
3	0.866	2.634	5.07 × 10^8^	4.86 × 10^8^
4	1.750	0.500	7.28 × 10^8^	7.63 × 10^8^
5	1.750	1.750	5.19 × 10^8^	5.45 × 10^8^
6	1.750	1.750	5.47 × 10^8^	5.45 × 10^8^
7	1.750	1.750	4.45 × 10^8^	5.45 × 10^8^
8	1.750	1.750	5.99 × 10^8^	5.45 × 10^8^
9	1.750	1.750	6.14 × 10^8^	5.45 × 10^8^
10	1.750	3.000	7.89 × 10^8^	7.83 × 10^8^
11	2.634	0.866	7.91 × 10^8^	7.84 × 10^8^
12	2.634	2.634	7.28 × 10^8^	7.63 × 10^8^
13	3.000	1.750	6.92 × 10^8^	6.75 × 10^8^

**Table 4 jof-08-01037-t004:** Observed and predicted values for P_aib_ production for each experiment for the validation of the central composite model that evaluates the concentrations of Aib and *F. oxysporum*.

Trial	Treatment (g/L)		P_aib_ Production (cps)	
	Aib	*F. oxysporum*	Observed	Predicted
1	2.190	1.750	8.79 × 10^8^	6.11 × 10^8^
2	1.300	1.750	6.21 × 10^8^	4.54 × 10^8^
3	1.750	1.300	6.32 × 10^8^	5.71 × 10^8^
4	1.750	2.190	4.22 × 10^8^	5.76 × 10^8^
5	2.190	2.190	5.72 × 10^8^	6.37 × 10^8^
6	2.190	1.300	5.29 × 10^8^	6.43 × 10^8^
7	1.300	2.190	6.14 × 10^8^	4.91 × 10^8^
8	1.300	1.300	6.26 × 10^8^	4.74 × 10^8^
9	2.634	0.866	6.18 × 10^8^	7.84 × 10^8^
10	2.634	0.866	5.25 × 10^8^	7.84 × 10^8^

**Table 5 jof-08-01037-t005:** Amino acid sequences of the P_aib_ produced by *T. asperellum*. N: N-terminal modification; Ac: acetylation; Lxx: leucine/isoleucine; Vxx: valine/isovaline.

P_aib_	*m*/*z*	N	1	2	3	4	5	6	7	8	9	10	11	12	13	14	15	16	17	18
Trichotoxin T5D2 ^1^	1676	Ac	Aib	Gly	Aib	Lxx	Aib	Gln	Aib	Aib	Ala	Ala	Ala	Aib	Pro	Lxx	Aib	Aib	Glu	Valol
Trichotoxin 1690	1691	Ac	Aib	Gly	Aib	Lxx	Aib	Gln	Aib	Aib	Ala	Ala	Ala	Aib	Pro	Lxx	Aib	Vxx	Glu	Valol
Trichotoxin 1703A ^3^	1704	Ac	Aib	Gly	Aib	Lxx	Aib	Gln	Aib	Aib	Aib	Ala	Ala	Aib	Pro	Lxx	Aib	Vxx	Gln	Valol
Trichotoxin A-40 ^2^	1705	Ac	Aib	Gly	Aib	Lxx	Aib	Gln	Aib	Aib	Aib	Ala	Aib	Aib	Pro	Lxx	Aib	Aib	Glu	Valol
Trichotoxin 1717A ^3^	1718	Ac	Aib	Gly	Aib	Lxx	Aib	Gln	Aib	Aib	Aib	Ala	Aib	Aib	Pro	Lxx	Aib	Vxx	Gln	Valol
Trichotoxin A-50 G ^1^	1726	Ac	Aib	Gly	Aib	Lxx	Aib	Gln	Aib	Aib	Aib	Ala	Ala	Aib	Pro	Lxx	Aib	Vxx	Gln	Valol

Source: ^1^ [57], ^2^ [58], ^3^ [53].

## Data Availability

Not applicable.

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
