# Peer review of "Peptaibol Production and Characterization from Trichoderma asperellum and Their Action as Biofungicide"

_jof, 2022, doi:10.3390/jof8101037_

Round 1

Reviewer 1 Report

Comments to the Author

This manuscript describes the production, characterization and fungicidal effects of Peptaibols from Trichoderma  asperellum. Authors optimized the production of Paib in Trichoderma, and tested the antagonistic ability to pathogenetic fungi on plates and tomatoes. These works were detailed and interesting, deepened the understanding of peptides in Trichoderma. 

I have some concerns in the current state of this manuscript:

-Figure 1 and Figure 2. Image content needs to be enriched, such as adding saliency analysis and error lines.

- When measuring Paib Production, the sampling time needs to be optimized. It is recommended to take samples at a fixed time interval.

- Enrich the section of the article on the antimicrobial activity of Paib, including the determination of MIC, MFC and IC50,  it is dangerous to directly test antimicrobial activity of Paib at 800 ug.ml-1(Line 200).

-Line 109: ‘C. gloeosporioides, F. oxysporum and B. cinerea” change to“C. gloeosporioides, F. oxysporum and B. cinerea’.

-Line 29:”similai to” should be changed as ‘is similar to’.

-Line 43 :”consist in “should be changed as ‘consist of’.

Line 94. ‘T. asperellum’ should be italic.  

Line 109.  The genus and species name should be italic. 

-Line 239:”lower” should be changed as”less than”.

- Figure 2:  Typographical error.

Line 303.  ‘sensing of molecule such as oligopeptides and oligochitosaccharides from the cell membrane of other fungi’.  How did author keep the stabilization of components of cell debris in preparing and autoclave the media?

Line 470.  The concentration of clotrimazole should be added here. Different concentration of fungicides has different effect.

Line 478.  Unify Paib or Paib in the whole paper. 

Reviewer 2 Report

  • Brief summary: The manuscript focus on the Peptaboils (Paib) produced by Trichoderma asperellum and their activities as biofungicides. In particular authors investigated the optimization of Paib production through manipulation of different factors including the carbon sources (sugars), additive amino acids and cell debris as elicitors. Paib produced according to the optimized method were purified, sequenced and identified by HPLC coupled to mass spectrometry. A Paib prototype was prepared with the extracts obtained from the optimized fermentations and the biological activity against fungi has been evaluated in vitro and in vivo. Furthermore the structure of different pathogenic fungi treated with Paib was observed through electron microscopy and alterations of treated spores and hyphae were demonstrated in comparison to control.
  • General concept comments: The manuscript is generally clear and well written and structured in a clear manner. In my opinion the introduction is quite short and the different paragraphs could be more smoothly connected but the essential information are reported.                                                 The experimental design it is well build, the experiments are properly planned and according to the information provided in methods section experiments are reproducible.                                        Data are well reported in Tables and clearly graphically represented in Figures and are consistently and properly interpreted in the manuscript. I noticed the absence of statistical letters in the figures, it would be easier for the readers to have statistical analysis reported graphically (e.g. as letters) in the figures.                                                                                                                          Conclusions are consistent with results and in line with data presented.

Specific comment: 

Line 24: "Aib" this abbreviation is used here for the first time, it should be written in full and abbreviation between brackets.

Line 27: "in vitro and in vivo" should be in italics

Line 109: "the pathogenic fungi C. gloeosporioides, F. oxysporum and B. cinerea " Names of fungi should be in italics.

Line 282: "..in T. longibrachiatum SMF2" If I am not wrong this name is for the first time mentioned in the manuscript so it should be written in full as Trichoderma longibrachiatum.

Figure 2 -> I was wondering why the statistical differences that are described in the text (line 310-312 and 321-322) are not graphically represented in Figure 2.

Line 337: Space missing between sentences "...(Paib Database [52]).These..."

Similarly to Figure 2 also in Figure 3 -> the significant differences described in line 345-347 are not graphically represented in Figure 3.

Concerning Figures 2 and 3 I think it would be better to be able to determine the statistical differences among the different objects tested just looking at the figures. The figures should be totally self explanatory.

Line 465-467: "The prototype demonstrated to be more effective against C. gloeosporioides with a GH of 92.2%. The GH for B. cinerea, A. alternata and F. oxysporum were 74.29, 58.4 and 36.2%, respectively." Growth inhibition values are reported in the text but it is not specified at which time point of the experiment these values correspond. The mycelial growth was assessed at different time points (days after incubation) and it is not clear which data point has been used to calculate GH. Please, clarify.

Also in Figure 5 the statistical analysis are not reported in the figure but only partially mentioned in the text (Line 468-469).

Reviewer 3 Report

Trichoderma asperellum and their Action as Biofungicide IS  very good title and interested to readers, thanks.

I completely love the figures, look presented and clear.

Reviewer 4 Report

The authors describe in their article the production of peptaibols by Trichoderma asperellum under different elicitation conditions consisting of the use of different carbon sources, the addition of several amino acids to the culture media and cell debris of different pathogenic fungi cultures. Using a model, they determine the best conditions for the production of peptaibols and use an extract prepared under these conditions to evaluate the in vivo antifungal activity as well as the effects in the morphology of pathogenic fungi treated with this extract.

Although there is some value in the research performed, this review cannot recommend the publication of the article in its current state due to the following reasons:

1)   No information is included in the article regarding the isolation and taxonomic identification of the T. asperellum strains used in the research.

2)   The authors mention in their article the purification of peptaibols but their purification seems to be limited to the generation of a crude extract from the fungal cultures. In all the analytical studies leading to the determination of the best production conditions a solid phase extraction using C18 reversed phase silica gel has been performed in which, after water washing, the column was eluted with ethanol 96 % v/v. For the determination of biological activities after optimization of the fermentation parameters, what the authorsrefer as prototype is a simple ethyl acetate extract with any further purification. In both cases the extracts will contain for sure other components whose presence has been completely overlooked. Some of these components might also contribute to the antifungal activity observed and their presence must therefore be accounted and evaluated.

3)   In this sense, the analytical work performed seems to be limited to a relative quantification of peptaibols production under the different conditions tested, without a real absolute quantification of the presence of these compounds in the extract to confirm the relevance of the experiments performed when compared to previous works describing the presence of the same compounds in cultures of the same fungus.

4)   The biological activity measured for the prototype prepared is somehow predictable based on previous reports. The relevance of the findings in this article, if any, must be highlighted with respect to previous studies. In this sense, a real purification of a peptaibol fraction would have been desired in order to evaluate their real contribution to the antifungal activity observed in the absence of other components of the extracts.

5)   Finally, a more detailed description of the effects observed by SEM would have been desirable. In this sense, please note that the identification of the treated and untreated versions of the fungal cultures in the caption of Figure 7 seems to be wrong.

Round 2

Reviewer 4 Report

I regret to say that my major concerns raised in the first revision of the article still remain unanswered:

1) When asking about the isolation of the fungus I do not refer to a single line stating that the fungus was isolated from agricultural soil but to the experimental procedure used in its isolation.

2)The authors assume but do not demonstrate that the extract could contain other minor components contributing to the fungal activity. It is striking that an article describing differences the optimization conditions for the production of peptaibols does not show a single analytical LC/UV/MS profile of the extracts obtained to evaluate the significance of the peptaibols versus the rest of components present in the extracts.

3) What the authors refer as quantification is not a real quantification of the content of peptaibols in the assays. They assume that the fungal extract added to the prototype only contains peptaibols, which is not true. In order to do a real quantification, standards are needed, an to obtain those standards, a purification of the compounds present in the extract is required.

I therefore must stand by my previous decision of not recommending the article for publication